# Solid Oxide Fuel Cells with Magnetron Sputtered Single-Layer SDC and Multilayer SDC/YSZ/SDC Electrolytes

**DOI:** 10.3390/membranes13060585

**Published:** 2023-06-05

**Authors:** Andrey Solovyev, Anna Shipilova, Egor Smolyanskiy

**Affiliations:** 1Institute of High Current Electronics SB RAS, 634055 Tomsk, Russia; lassie2@yandex.ru; 2Department of Physics, National Research Tomsk State University, 634050 Tomsk, Russia; 3School of Advanced Manufacturing Technologies, National Research Tomsk Polytechnic University, 634050 Tomsk, Russia; smolianskyea@ya.ru

**Keywords:** solid oxide fuel cell, thin-film electrolyte, anode-supported cell, doped CeO_2_, magnetron sputtering, YSZ, SDC

## Abstract

Samarium-doped ceria (SDC) is considered as an alternative electrolyte material for intermediate-temperature solid oxide fuel cells (IT-SOFCs) because its conductivity is higher than that of commonly used yttria-stabilized zirconia (YSZ). The paper compares the properties of anode-supported SOFCs with magnetron sputtered single-layer SDC and multilayer SDC/YSZ/SDC thin-film electrolyte, with the YSZ blocking layer 0.5, 1, and 1.5 μm thick. The thickness of the upper and lower SDC layers of the multilayer electrolyte are constant and amount to 3 and 1 μm, respectively. The thickness of single-layer SDC electrolyte is 5.5 μm. The SOFC performance is studied by measuring current–voltage characteristics and impedance spectra in the range of 500–800 °C. X-ray diffraction and scanning electron microscopy are used to investigate the structure of the deposited electrolyte and other fuel cell layers. SOFCs with the single-layer SDC electrolyte show the best performance at 650 °C. At this temperature, open circuit voltage and maximum power density are 0.8 V and 651 mW/cm^2^, respectively. The formation of the SDC electrolyte with the YSZ blocking layer improves the open circuit voltage up to 1.1 V and increases the maximum power density at the temperatures over 600 °C. It is shown that the optimal thickness of the YSZ blocking layer is 1 µm. The fuel cell with the multilayer SDC/YSZ/SDC electrolyte, with the layer thicknesses of 3/1/1 µm, has the maximum power density of 2263 and 1132 mW/cm^2^ at 800 and 650 °C, respectively.

## 1. Introduction

In recent years, research related to lowering the operating temperature of solid oxide fuel cells has become very popular. Development of intermediate-temperature solid oxide fuel cells (IT-SOFC) induces a number of advantages. Such conditions widen opportunities for choosing materials to manufacture IT-SOFCs, improving their durability and increasing attractiveness to consumers [1]. However, the lower operating temperature causes many problems. The main problems are associated with growing ohmic and polarization resistances of the fuel cell. The increase in the area-specific resistance (ASR) of fuel cells is conditioned by thermal activation properties of components [2,3]. Application of mixed ionic–electronic conducting materials as the cathode leads to an acceleration of the oxygen reduction reaction (ORR) and improvement in the fuel cell performance [4,5,6,7]. As for the electrolyte, its thickness must be as thin as possible. Also, materials with high ionic conductivity are preferably used for the electrolyte to increase the SOFCs’ efficiency [3,8,9,10].

Yttria-stabilized zirconia (YSZ) is a commonly used electrolyte material. It demonstrates good ionic conductivity at high (above 800 °C) temperature and high ionic transference number in both oxidizing and reducing atmospheres. The YSZ use in IT-SOFCs is, however, difficult in the operating temperature range of 600 to 800 °C, since YSZ ionic conductivity drastically lowers with decreasing operating temperature. Such conditions require alternative thin-film electrolytes which have higher oxygen-ionic conductivity than YSZ. Candidates for such solid electrolytes include cerium oxide (CeO_2_) doped with rare earth or alkaline elements. According to [11], Gd_2_O_3_ and Sm_2_O_3_ are most-often used as dopant impurities. GDC and SDC show the highest ionic conductivity among cerium-based oxides due to relatively equal ionic radii of dopants Sm^3+^, Gd^3+^, and Ce^4+^ [12,13]. However, doped ceria has one drawback associated with its oxidation–reduction instability. In a hydrogen atmosphere, at high (above 650 °C) operating temperatures, cerium partially reduces, viz., Ce (IV) → Ce (III), which leads to the appearance of mixed ionic–electronic conduction. As a result, the open circuit voltage (OCV) of the fuel cell reduces, and the increased electronic conductivity can lead to the short-circuit problem in the SOFC [14].

Different approaches are used to improve the stability of doped CeO_2_ in a reducing atmosphere and further enhance its conductivity. The focus of recent research is, for example, on triple-doped ceria-based materials such as Ce_1−*x*_(La_1/3_Sm_1/3_Gd_1/3_)*_x_*O_2−δ_ (LSG) [15,16], Ce_0.8_Sm_0.12_Pr_0.08_O_2−δ_ [17], Ce_0.8_Nd_0.18_Ca_0.02_O_1.89_ [18], and Ce_0.83_Dy_0.14_Ca_0.03_O_1.90_ [19]. GDC co-doped with rare earth elements (CeO_2_–8Gd_2_O_3_–2Eu_2_O_3_ (Eu–GDC) and CeO_2_–8Gd_2_O_3_–2Nd_2_O_3_ (Nd–GDC)) are synthesized and studied in [20]. Such multicomponent compositions are characterized by an enhanced total ionic conductivity compared to the ionic conductivity of single-doped ceria materials. The literature shows detailed studies of the microstructure, phase composition, and conductivity of these materials. However, these compositions are rather complex and require detailed study of their stability and efficiency, including their application as electrolyte in a real device, namely, an SOFC.

Mishima et al. [21] showed that the formation of a YSZ- and SDC-containing composite electrolyte improved the SOFC output voltage. The OCV of the fuel cell with biphasic electrolyte was significantly higher (~1.1 V) than that of the SDC electrolyte (0.89 V). Incorporation of the SDC grains into the YSZ matrix proved to be effective for reducing the overall internal resistance of the cell. Raghvendra and Singh [22] demonstrated a significant decrease in the electrode polarization when adding the SDC phase into the YSZ electrolyte, and the electrical conductivity of YSZ–SDC was higher when compared with YSZ electrolyte at 400–700 °C. Thus, after the formation of binary composite electrolytes, the SOFC performance could be expected to be higher than in SOFCs with single SDC or YSZ electrolytes.

One more approach to increase the doped CeO_2_ stability in a reducing atmosphere was described in [23,24,25,26], which demonstrated the application of a thin YSZ blocking layer between anodes and CeO_2_-based electrolyte. Eguchi et al. [27] found that OCV of the fuel cell with YSZ-coated SDC approached the theoretical OCV value of the fuel cell with YSZ electrolyte. The maximum power density of fuel cell with YSZ-coated SDC electrolyte was 320 mW/cm^2^ at 800 °C, which is more than two times higher than that (120 mW/cm^2^) for the SOFC with the single-layer YSZ electrolyte at the same temperature.

The high ionic conductivity and low reactivity with electrode materials of CeO_2_-based electrolytes [28] create a promising approach for electrolyte fabrication in the form of a multilayer thin-film structure. For example, the application of ZrO_2_-based thin-film electrolyte has been performed between two layers of doped CeO_2_, viz., GDC/ScSZ/GDC [29] or GDC/YSZ/GDC [26,30]. Remarkable power densities of 2580 mW/cm^2^ at 800 °C and 2100 mW/cm^2^ at 650 °C were obtained in [26,30] for SOFCs with the GDC/YSZ/GDC thin-film electrolyte. This multilayer structure based on YSZ and SDC electrolytes has not been implemented until the present day. Considering such striking aspects of ceria-based oxides as their low reactivity with the cathode materials on the one hand and promoting activity for the anodic reaction of nickel on the other [27], we believe that an electrolyte structure similar to SDC/YSZ/SDC can be effective to improve the fuel cell performance.

According to Liang et al. [31], thin-film electrolytes can be successfully fabricated by spin coating, dip coating, atomic layer deposition, and magnetron sputtering. Several publications [32,33] focus on the fabrication of GDC thin films by the magnetron sputtering method. There is, however, no information in the literature about magnetron sputtering of SDC thin films. Only Yoo [34] has fabricated Sm_0.2_Ce_0.8_O_1.9_ thin films using the high-frequency magnetron sputtering method, at 0.10 to 0.15 µm/h deposition rate. Single SOFCs with an SDC electrolyte 1 or 2 µm thick displayed 0.85 V OCV at 700 °C, although there were no measurement results of current–voltage characteristics in his work. SOFCs with the bilayer ScSZ/SDC (with layer thicknesses of 1.2/1.5 µm) electrolyte demonstrated OCV and power density values of 1.02 V and 360 mW/cm^2^ at 600 °C, respectively.

SOFCs with 1 µm thick SDC electrolyte, prepared in [35] by modified sol–gel route, were characterized by 0.83 OCV at 500 °C and maximum power densities of 59.6, 121.9, and 133.8 mW/cm^2^ at 500, 600, and 700 °C, respectively. Yoo et al. [36] fabricated an SDC electrolyte 30 µm thick by using suspension plasma spray for tubular SOFCs with metal base. Those SOFCs had 0.88 OCV and 165 mW/cm^2^ power density at 0.6 V and 600 °C. Ding et al. [37] fabricated SDC films by pulsed laser deposition (PLD). The conductivity of films was 0.015 S/cm at 600 °C, which was almost 3 times higher than sol–gel-prepared films [38]. It was also noted that the film formation temperature when using PLD (600–700 °C) was considerably lower than the formation temperature when using spray coating (1300 °C) [39], electrophoretic deposition (1400 °C) [40], screen-printing [41], and others. Thus, a significant advantage of PLD is a much lower process temperature, which allows for avoidance of the potential chemical reactions at the electrolyte–electrode interface at high operating temperatures of fuel cells.

The literature review shows the absence of works in the area of SDC deposition by reactive magnetron sputtering of Ce–Sm metal targets, even though this technique provides high deposition rates as compared to high-frequency magnetron sputtering of oxide targets and is more suitable for industrial applications.

The aim of this work is to investigate the structure and properties of the SDC thin-film electrolyte deposited by reactive magnetron sputtering. The latter is also used to create the multilayer SDC/YSZ/SDC electrolyte structure to improve the OCV and efficiency of the SOFC. The microstructure and phase composition of the SDC electrolyte are studied by also using synchrotron radiation. The cell performance is investigated by current–voltage (I–U) curves and using impedance spectroscopy. The performance of SOFCs with single-layer SDC and multilayer SDC/YSZ/SDC electrolytes is compared and discussed.

## 2. Materials and Methods

### 2.1. SDC and YSZ Layer Deposition

SDC (Ce_0.8_Sm_0.2_O_2−δ_) and YSZ ((ZrO_2_)_0.92_:Y_2_O_3_)_0.08_) electrolytes were deposited by reactive magnetron sputtering of Ce–Sm (80/20 at.%) and Zr–Y (85/15 at.%) metal targets produced by Advanced Ceramic Esoterica Store (Beijing, China) and Girmet (Moscow, Russia), respectively. NiO/YSZ anodes (Kceracell Co., Chungcheongnam-Do, Republic of Korea) of diameter 20 mm and 0.7 mm thick were used as substrates. Prior to the coating deposition, the substrates were cleaned ultrasonically in isopropyl alcohol and then two times in deionized water. Afterwards, substrates were placed on a holder with a resistive heater. The vacuum chamber was evacuated to a residual pressure of 6 × 10^−5^ Torr, and substrates were heated up to 300 °C, which was kept during the deposition process. After that, argon was fed into the chamber at a rate of 110 sccm, and substrates were cleaned with an ion source with closed electron drift at 1.5 kV and 25–30 mA for 10 min. Then, oxygen was fed into the chamber, and the resulting Ar–O_2_ atmosphere was used for the electrolyte deposition.

This deposition process is schematically illustrated in Figure 1. During the process, the substrate holder was turned to the respective magnetron such that the substrate and target were parallel to each other and situated at a distance of 75 mm from each other. The multilayer SDC/YSZ/SDC electrolyte was sputtered in one technological cycle without vacuum loss in the chamber.

The SDC electrolyte was deposited at a process pressure of 8 × 10^−3^ torr. The gas flow rates were 100 and 50 sccm for argon and oxygen, respectively. The discharge power was constant and amounted to 0.5 kW. The deposition rate was 1 µm/h. The oxygen flow rate for the YSZ layer deposition was reduced to 40 sccm. At the discharge power of 0.5 kW, the deposition rate was 0.5 µm/h.

Magnetron sputtering systems were supplied by the pulsed power supply APEL-M-12DU-symmetric (Applied Electronics, Tomsk, Russia), generating 50 kHz frequency asymmetric bipolar rectangular pulses.

### 2.2. Preparation of Single Fuel Cells

Screen-printing was used to create a two-layer cathode on half-cells comprising the anode with single- or multilayer electrolytes for electrochemical testing. The La_0.6_Sr_0.4_Co_0.2_Fe_0.8_O_3_–Ce_0.9_Gd_0.1_O_2_ (LSCF–GDC, 50:50 wt.%) paste (CERA-FC Co., Geumsan, Korea) was used to create the cathode functional layer. Two layers of the LSCF–GDC paste were applied to the electrolyte surface. Each layer was dried at 100 °C for 5 min, and then they were sintered at 1050 °C for 2 h. Prior to electrochemical testing, the cathode contact layer made of the La_0.6_Sr_0.4_CoO_3_ (LSC) paste (CERA-FC Co., Geumsan, Republic of Korea) was printed on the LSCF–GDC cathode functional layer. The LSC cathode contact layer was sintered in situ during the fuel cell testing at 800 °C. The cathode area was 1 cm^2^.

### 2.3. Investigation of Single Fuel Cells

The anode-supported fuel cell electrical characterization was provided by a ProboStat^TM^ system (NorECs, Oslo, Norway) in the temperature range of 500–700 °C for the fuel cell with the SDC electrolyte, and 500–800 °C for the fuel cell with the SDC/YSZ/SDC electrolyte. The fuel cell placed in the test station was heated up to 800 °C for 2.5 h, kept at this temperature for 1 h, and then the current–voltage and impedance characteristics were measured. Glass sealing was used to separate gas spaces. Working gases were dry hydrogen and air consumed at 18 and 300 mL/min flow rates, respectively, for the single-layer SDC electrolyte, and at 105 and 400 mL/min flow rates for the multilayer SDC/YSZ/SDC electrolyte. Pt wires and Ag mesh were used for current collection from the anode and cathode. I–U characteristics and impedance spectra were measured by a P-150S potentiostat/galvanostat and a Z-500P impedancemeter (Elins, Moscow, Russia). The impedance was measured in the frequency range of 0.1–5 × 10^5^ Hz, and AC signal of 5 mV in amplitude was obtained under open circuit conditions.

An Apreo 2S Scanning Electron Microscope (SEM) (Thermo Fisher Scientific, Waltham, MA, USA) was used to study the electrolyte microstructure and thickness measurements.

The phase composition of electrolytes was investigated on a Shimadzu XRD-6000 Diffractometer (Kyoto, Japan) using Cu-*K*_α_ radiation operated at 40 kV and 30 mA. The analysis of the phase composition, coherent scattering regions, and microstresses (∆*d*/*d*) was performed using PDF4+ databases and PowderCell 2.4.

In addition to the conventional X-ray diffraction (XRD) analysis, synchrotron radiation (SR) was used to study the phase composition of the SDC electrolyte. The XRD analysis was conducted for the SR beam of the VEPP-3 storage ring, in Siberian Synchrotron and Terahertz Radiation Centre of the Budker Institute of Nuclear Physics SB RAS, Novosibirsk, Russia. SDC electrolyte was heated from 30 to 1300 °C at a heating rate of 15 °C/min. The operating wavelength was 0.172 nm. The diffraction angle was recalculated to the angle at 0.1541 nm wavelength (Cu-K_α_ radiation) for comparing the obtained results with those measured by a commercial X-ray diffraction apparatus.

## 3. Results and Discussion

### 3.1. Fuel Cell with the Single-Layer SDC Electrolyte

Figure 2 shows cross-sectional SEM images of the magnetron sputtered SDC electrolyte obtained on the anode support after deposition and subsequent annealing at 1000 °C in air. It is known that high-temperature annealing provides the growth in crystallinity of magnetron sputtered YSZ and GDC layers [42,43]. The 5 µm thick electrolyte has good adhesion to the substrate. One can see that after 1000 °C annealing, the coating microstructure does not change.

Figure 3 presents XRD patterns of SDC layers 5 µm thick after the deposition and annealing at 900, 1000, and 1100 °C in air. The results of the XRD analysis are given in Table 1. The as-deposited SDC layer has the cubic crystal structure of fluorite with the space group *Fm-3m* (225) (the PDF file number 75-0158) and (111) preferred crystallographic orientation. The (111) and (222) peaks shift toward lower 2θ angles as compared to reference values. This indicates the presence of compressive stresses in the film. After annealing, the peak positions shift toward large angles, which indicates reduced residual stresses. After 900 °C annealing, the peak (331) is observed. When the annealing temperature rises, the lattice parameter reduces, the size of coherent scattering region (CSR) increases, and microstresses (∆*d*/*d*) decrease.

In addition to the conventional XRD analysis, SR radiation was used to study the phase composition of the SDC layer during the process of heating it to 1300 °C. According to Figure 4a, the as-deposited layer shows (111) and (220) peaks within the angular range of 2θ = 27–54°. The peak (200), matching 2θ = 32.9°, is not detected. The peaks widen and shift towards the low-angle region that, along with the absence of the peak (200), indicates a disordered structure or compressive stresses. Weak reflections are detected for the NiO/YSZ substrate.

A series of synchrotron X-ray diffraction (SXRD) patterns of the SDC layer is presented in Figure 4b as the intensity projection on the diffraction angle–time plane. These patterns are obtained during heating to 1300 °C in air and subsequent cooling down to room temperature. One can see that, during the heating process, the SDC reflections stop shifting to the left at ~770 °C due to thermal expansion, and their position is almost constant until 1300 °C. At the same time, the peak width narrows and the intensity grows. This is due to defect annealing of the electrolyte structure formed in non-equilibrium conditions and the structure saturation with oxygen from air.

After heating and cooling to room temperature, the peak position matches stoichiometric material, i.e., Sm_0.2_Ce_0.8_O_1.9_. As presented in Figure 4a, the peak width reduces by about two times and, concurrently, the peak intensity increases by two times; thus, the total intensity does not change.

### 3.2. Fuel Cell with Multilayer SDC/YSZ/SDC Electrolyte

The SEM image in Figure 5a shows the microstructure of the multilayer SDC/YSZ/SDC electrolyte with the layer thicknesses of 3, 1, and 1 µm. This image is obtained after the cell testing in the mode, which combines signals of secondary and backscattered electrons in the ratio 50:50. This imaging mode allows detection of the phase contrast and details of the coating morphology in one image. As can be seen in Figure 5a, the electrolyte has a dense structure and good adhesion, both between electrolyte layers and to the anode. The YSZ blocking layer is dense, without defects such as pores, cracks, and delamination. The formation of this dense defect-free structure is the important condition for blocking the electrolyte electronic conductivity. The XRD pattern in Figure 5b shows the presence of SDC and YSZ phases with cubic crystal structure in the multilayer SDC/YSZ/SDC electrolyte.

### 3.3. Single Fuel Cell Testing

OCV temperature dependences for SOFCs with single-layer SDC and multilayer SDC/YSZ/SDC electrolytes are presented in Figure 6. OCV of the SOFC with the single-layer electrolyte (5.5 µm thick) does not surpass 0.85 V at 500 °C and gradually lowers to 0.77 V as operating temperature increases up to 700 °C. For comparison, this figure shows OCV values of SOFCs with the single-layer SDC electrolyte having thicknesses of 0.5 µm [35], 10 µm [44], and 400 µm [45]. OCV depends not only on temperature, but also thickness of the SDC electrolyte, although OCV of anode-supported fuel cells with the electrolyte having 0.5 to 10 µm thickness does not strongly depend on the thickness. However, the use of electrolyte-supported fuel cells with electrolyte thickness spanning several hundred micrometers provides 0.9 to 0.94 OCV. An explanation for the dependence of OCV on the thickness of the SDC electrolyte may be the smaller number of defects in the thicker electrolyte which pass through the entire thickness of the electrolyte. This leads to a higher gas impermeability in thick electrolytes. It may also be that the partial oxygen pressure distribution inside the electrolyte depends on its thickness. Nevertheless, OCV of the SOFC with the magnetron sputtered SDC electrolyte indicates very low gas permeation through the electrolyte layer, which is in good agreement with cross-sectional SEM observation of the electrolyte.

The addition of the YSZ thin blocking layer to the electrolyte structure leads to a significant OCV growth. Testing of cells with the YSZ blocking layer 0.5, 1, and 1.5 µm thick shows that, over the whole temperature range (500–800 °C), OCV of fuel cells with the multilayer SDC/YSZ/SDC electrolyte exceeds 1 V. The highest OCV is observed for fuel cells with blocking layers 1 and 1.5 µm thick. In the range of 500–700 °C, OCV of these cells exceeds 1.1 V, i.e., close to theoretical. These data prove that the addition of the YSZ thin blocking layer in the SDC electrolyte effectively blocks the leakage current, which causes the OCV drop at high temperatures. The use of YSZ blocking layer less than 0.5 µm thick does not provide an effective blocking of the leakage current.

Table 2 presents the performance of fuel cells with single-layer SDC and multilayer SDC/YSZ/SDC electrolytes with different thickness of the YSZ blocking layer. Figure 7 shows I–U and current–power (I–P) curves of fuel cells with single-layer SDC electrolyte and multilayer SDC/YSZ/SDC electrolytes with the YSZ blocking layer 1 μm thick. The highest power density (651 mW/cm^2^) for the SOFC with the single-layer SDC electrolyte was obtained at 650 °C (Figure 7a). With the temperature decrease, the cell power density reduces due to the lower ionic conductivity of the electrolyte and higher activation losses on electrodes. The temperature growth up to 700 °C also causes the power reduction, which is probably associated with the higher electronic conductivity of the SDC electrolyte. This is also evidenced by a decrease in the OCV of this cell. The power density also strongly depends on the hydrogen flow rate; its maximum values are observed at a rate of 18 mL/min. When the latter is higher, the power density decreases due to the growth in the electronic conductivity of the electrolyte and the OCV reduction.

As expected, the addition of the YSZ blocking layer to the SDC electrolyte leads to the SOFC power growth (Figure 7b). In this case, the highest power density is observed at much higher (105 mL/min) hydrogen flow rate, which indicates a decrease in the electronic conductivity in the electrolyte. Compared to the single-layer SDC electrolyte, the most significant performance improvement is observed at temperatures higher than 650 °C. At 600 °C and lower, the SOFC performance with single- and multilayer electrolytes does not differ. Among SOFCs with the multilayer electrolyte, the best properties are obtained for the SOFC with the YSZ blocking layer 1 µm thick. Its power density is 1438 mW/cm^2^ at 700 °C, i.e., 2 times higher than for the SOFC with the single-layer SDC electrolyte. The highest (2263 mW/cm^2^) power density is detected at 800 °C.

The area-specific resistances (ASR) calculated by the I–U curve slope at 700, 600, and 500 °C are, respectively, 0.163, 0.252, and 0.814 mΩ·cm^2^ for the SOFC with the SDC electrolyte, and 0.172, 0.275, and 0.838 mΩ·cm^2^ for the SOFC with the SDC/YSZ (1 μm)/SDC electrolyte, i.e., ASR values are almost the same. This means that the addition of the YSZ blocking layer 1 µm thick to the SDC electrolyte does not lead to a significant growth in the SOFC ohmic resistance. The similar behavior is described in [27] for the fuel cell with the thick SDC electrolyte coated by 2 µm thick YSZ film. However, the thickness growth of the YSZ blocking layer up to 1.5 µm (see Table 2) leads to faster performance degradation of the SOFC with decreasing temperature. This is particularly observed at temperatures below 650 °C, when the contribution of the ohmic resistance of the YSZ layer becomes more notable.

The impedance analysis is conducted to determine the contribution of the YSZ layer to ohmic (*R*_ohm_) and polarization (*R*_pol_) resistances. Figure 8 presents impedance spectra of SOFCs with single-layer SDC and multilayer SDC/YSZ (1 μm)/SDC electrolytes. The values of *R*_ohm_ are summarized in Table 3 for all fuel cells, as obtained from the impedance plots.

The SOFC ohmic resistance, as related to the resistance in the electrolyte, electrode, and their interface, is determined by the spectrum intercept with Re(*Z*) axis in the high-frequency region. Since SOFCs differ by the electrolyte composition only, *R*_ohm_ variation is attributed to the change in the electrolyte resistance. As can be seen from Table 3, the lowest *R*_ohm_ at low (<600 °C) temperatures belongs to the single-layer SDC electrolyte. At higher temperatures, the *R*_ohm_ of the cell with SDC/YSZ (1 μm)/SDC electrolyte becomes slightly smaller than that of a cell with a single layer electrolyte. In the latter, the thickness growth of the YSZ layer increases the ohmic resistance at any operating temperature.

The polarization resistance is determined for the SOFC with the SDC/YSZ (1 μm)/SDC electrolyte as a difference between intercepts with Re(*Z*) axis at low and high frequencies (see Table 4). This resistance is associated with activation and concentration losses in the fuel cell. Activation losses arise during the charge transfer from electronic and ionic conductors; this is also termed as charge transfer losses. Activation losses are typical for cathodes. These are attributed to the slow ORR kinetics, which involves the breakage of high-strength bonds in oxygen molecules [46]. Concentration losses occur when reacting molecules are consumed at a faster rate or when products are not quickly removed from reaction sites. 

According to Table 4, *R*_pol_ of both SOFCs grows with decreasing operating temperature, because the catalytic activity of electrodes is reduced. *R*_pol_ of the SOFC with the SDC/YSZ (1 μm)/SDC electrolyte is higher than with the single-layer electrolyte. This is because the electronic conductivity is absent and the fuel humidity in the SOFC with the multilayer electrolyte is low. For the latter, we hypothesize dependences of the impedance spectra on air and hydrogen flow rates at 600 and 750 °C and the OCV (see Figure 9). The air flow rate of 200 to 600 mL/min has no effect on the impedance spectra. As for the hydrogen flow rate, it significantly affects the SOFC polarization resistance. The impedance spectra are approximated by high- and low-frequency arcs. The high-frequency arc size does not depend on the hydrogen flow rate. In contrast, the low-frequency arc becomes larger when the hydrogen flow rate increases from 30 to 105 mL/min, in particular at 750 °C.

In Figure 9, one can see the increase in the total polarization resistance with increasing hydrogen flow rate, while the high-frequency intercept remains reasonably constant within the experimental error. This indicates that the ohmic resistance remains unchanged, as expected. It is known that the low-frequency arc observed at 0.1 to 1 Hz is attributed to the mass transfer resistance on the anode [47]. At low frequencies, the contribution from transport phenomena to the impedance dominates over contributions from other phenomena.

The total electrode polarization usually decreases as the hydrogen partial pressure grows [47,48,49]. In our case, the situation is reversed, which is explained by the use of dry hydrogen and the test station design, in which the anode gas chamber is not sealed and is, thus, directly connected with air outside the furnace. Therefore, at low rates of the dry hydrogen flow, the partial pressure of water in the chamber is rather high. When the flow rate of dry hydrogen grows, moisture in the chamber decreases, thereby causing the size growth of the low-frequency arc and, consequently, the electrode total polarization. The same dependence was observed by Leonide et al. [50] for the polarization resistance and partial pressure of water vapor (pH_2_O) in the fuel gas. The impedance spectra measured at different pH_2_O values were analyzed by the distribution of relaxation times approach. At higher pH_2_O, they observed intensified low-frequency (0.1 to 1 Hz) processes caused by diffusion losses on the anode substrate. 

## 4. Conclusions

The possibility of using magnetron sputtering was demonstrated for a successful fabrication of the thin film SDC electrolyte for IT-SOFCs. Magnetron sputtering provided the formation of the defect-free electrolyte, presenting a cubic phase and having good adhesion to the substrate. The OCV of the fuel cell with the SDC electrolyte 5.5 μm thick was about 0.8 V in the temperature range of 500 to 650 °C, which was then decreased to 0.77 V at 700 °C. The highest SOFC performance was observed at 650 °C, when its OCV and power density were 0.805 V and 651 mW/cm^2^, respectively. The SDC ability to reduce and, thus, promote the low OCV made its application ineffective as a thin-film electrolyte. The YSZ blocking layer incorporated into the electrolyte structure improved OCV and power density of fuel cells. It was shown that the presence of the blocking layer did not lead to ohmic resistance growth, and its optimal thickness was about 1 µm. At 650 °C, the SOFC with the SDC/YSZ (1 μm)/SDC electrolyte demonstrated 1.1 V and 1132 mW/cm^2^, i.e., 1.5 times higher than the values manifested by the SOFC with the single-layer electrolyte. The highest performance was observed for the fuel cell with the multilayer electrolyte at 800 °C; its OCV and power density were, respectively, 1.06 V and 2263 mW/cm^2^.

Despite the promising findings presented in this paper, further research is needed to test the long-term stability of multilayer electrolytes and their resistance to redox tests.

## Figures and Tables

**Figure 1 membranes-13-00585-f001:**
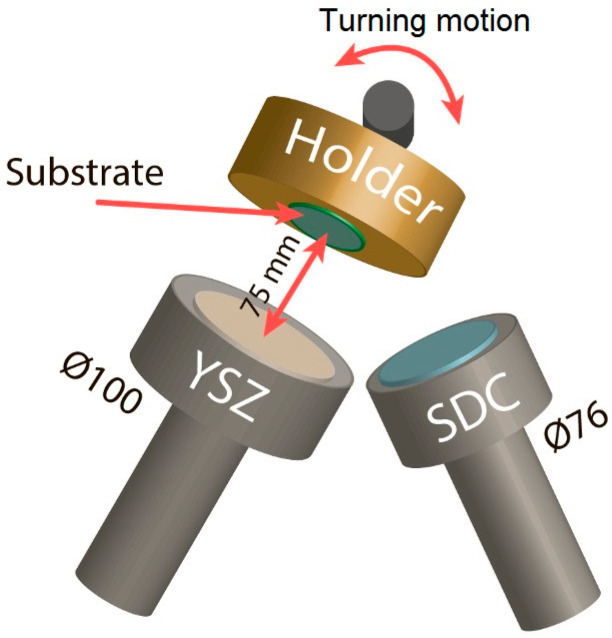
Schematic of SDC and YSZ layer deposition by magnetron sputtering.

**Figure 2 membranes-13-00585-f002:**
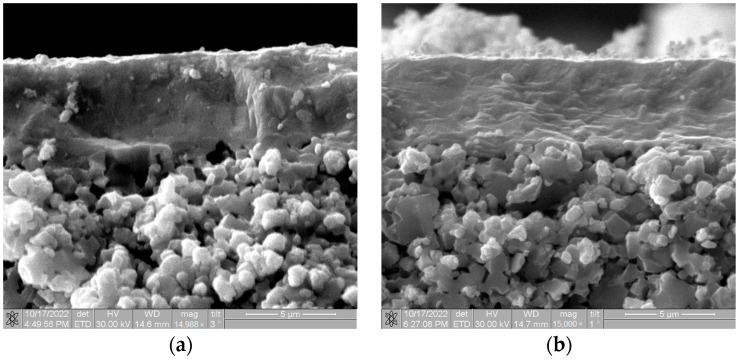
Cross-sectional SEM images of magnetron sputtered single-layer SDC electrolyte on the anode support: (**a**)—as-deposited, (**b**)—after 1000 °C annealing.

**Figure 3 membranes-13-00585-f003:**
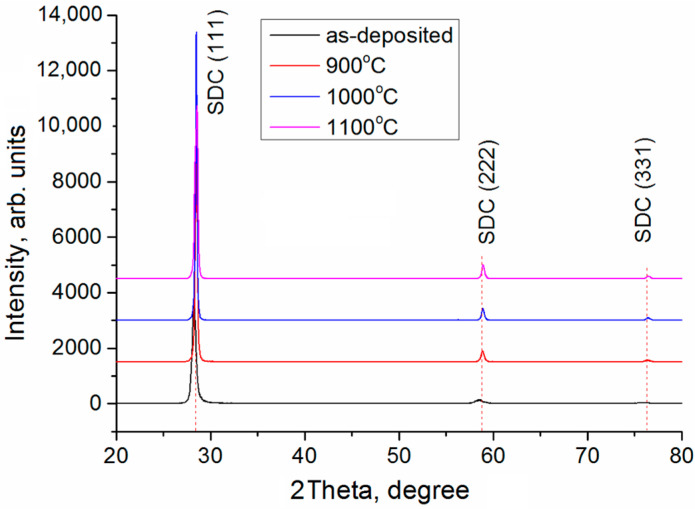
XRD patterns of single-layer SDC electrolyte annealed at different temperatures.

**Figure 4 membranes-13-00585-f004:**
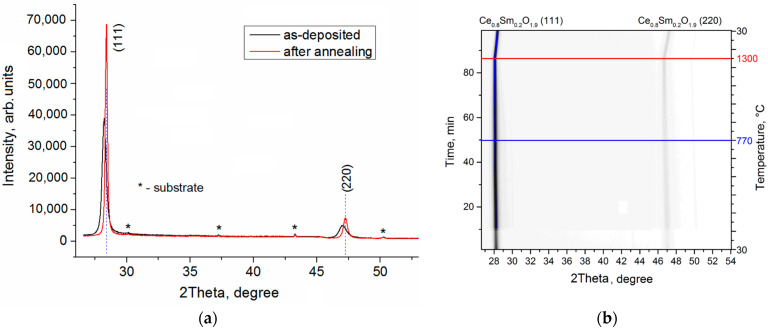
XRD patterns of SDC electrolyte on NiO/YSZ substrate: (**a**)—before and after 1300 °C annealing, (**b**)—during heating to 1300 °C and cooling to room temperature.

**Figure 5 membranes-13-00585-f005:**
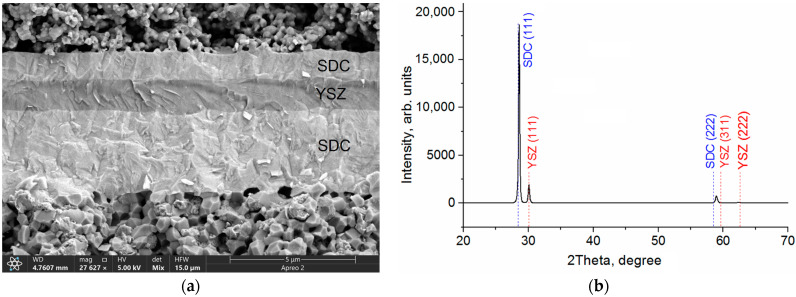
Cross-sectional SEM image (**a**) and XRD pattern (**b**) of multilayer SDC/YSZ/SDC electrolyte.

**Figure 6 membranes-13-00585-f006:**
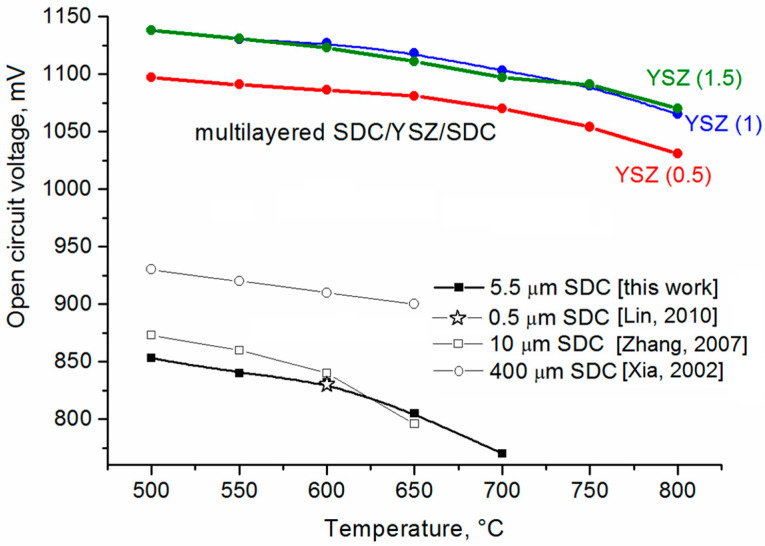
Temperature dependences of OCV of SOFCs with single-layer SDC electrolyte, reproduced from [35,44,45], and multilayer SDC/YSZ/SDC electrolyte with different thickness of YSZ layer.

**Figure 7 membranes-13-00585-f007:**
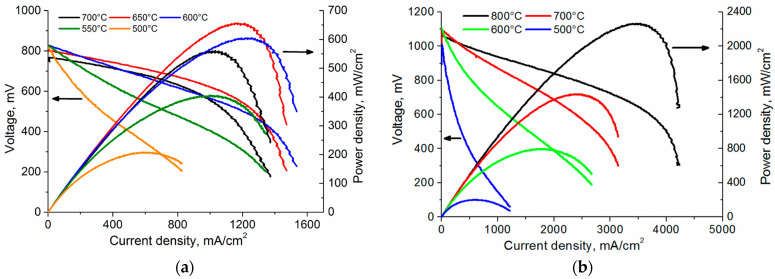
I–U and I–P curves of SOFCs with single-layer SDC (**a**) and multilayer SDC/YSZ(1 μm)/SDC (**b**) electrolytes.

**Figure 8 membranes-13-00585-f008:**
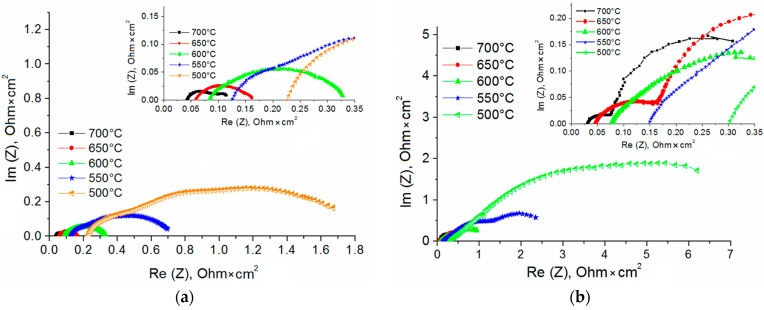
Impedance plots of SOFCs with single-layer SDC (**a**) and multilayer SDC/YSZ (1 μm)/SDC (**b**) electrolytes. Inserts show the high-frequency region.

**Figure 9 membranes-13-00585-f009:**
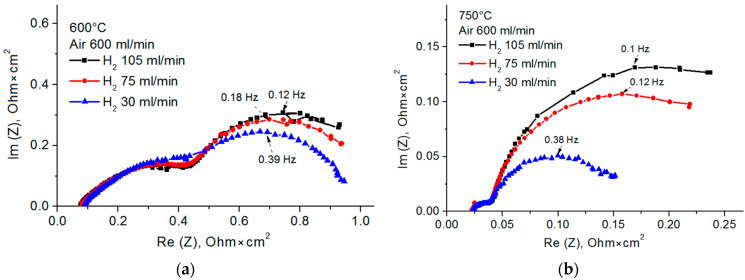
Impedance plots of SOFC with multilayer SDC/YSZ (1 μm)/SDC electrolyte measured at 600 °C (**a**) and 750 °C (**b**) at OCV and different hydrogen flow rates.

**Table 1 membranes-13-00585-t001:** Phase composition and structural parameters of SDC electrolyte.

SDC Films	Phases	Phase Content, vol.%	Lattice Parameters, Ǻ	CSR ^1^ Size, nm	∆*d*/*d* × 10^–3^
As-deposited	(Ce_0.8_Sm_0.2_)O_1.9_	100	*a* = 5.4495	39	1.1
Annealed at 900 °C	(Ce_0.8_Sm_0.2_)O_1.9_	100	*a* = 5.4375	129	0.3
Annealed at 1000 °C	(Ce_0.8_Sm_0.2_)O_1.9_	100	*a* = 5.4309	189	0.2
Annealed at 1100 °C	(Ce_0.8_Sm_0.2_)O_1.9_	100	*a* = 5.4348	92	0.3

^1^ CSR size and ∆*d*/*d* are calculated for peaks (111) and (222), respectively.

**Table 2 membranes-13-00585-t002:** Maximum power density of fuel cells with single-layer SDC and multilayer SDC/YSZ/SDC electrolytes at different temperatures.

Electrolyte Structures	Power Density (mW/cm^–2^)						
	800 °C	750 °C	700 °C	650 °C	600 °C	550 °C	500 °C
SDC (5.5 μm)	-	-	558	651	604	405	208
SDC/YSZ (0.5 μm)/SDC	1650	1523	1357	1080	782	469	213
SDC/YSZ (1 μm)/SDC	2263	1818	1438	1132	794	445	201
SDC/YSZ (1.5 μm)/SDC	1912	1450	1190	935	611	340	143

**Table 3 membranes-13-00585-t003:** Ohmic resistance of SOFCs with single-layer SDC and multilayer SDC/YSZ/SDC electrolytes at different temperatures.

Electrolyte Structure	Ohmic Resistance (Ω·cm^2^)						
	800 °C	750 °C	700 °C	650 °C	600 °C	550 °C	500 °C
SDC (5.5 μm)	-	-	0.04	0.05	0.08	0.12	0.22
SDC/YSZ (0.5 μm)/SDC	0.016	0.017	0.03	0.04	0.07	0.12	0.23
SDC/YSZ (1 μm)/SDC	0.017	0.02	0.03	0.04	0.07	0.14	0.29
SDC/YSZ (1.5 μm)/SDC	0.018	0.02	0.03	0.05	0.1	0.19	0.39

**Table 4 membranes-13-00585-t004:** Polarization resistance of SOFCs with single-layer SDC and multilayer SDC/YSZ (1 μm)/SDC electrolytes at different temperatures.

Electrolyte Structure	Polarization Resistance (Ω·cm^2^)						
	800 °C	750 °C	700 °C	650 °C	600 °C	550 °C	500 °C
SDC (5.5 μm)	-	-	0.08	0.1	0.23	0.61	1.6
SDC/YSZ (1 μm)/SDC	0.26	0.33	0.41	0.58	1.01	2.8	7.2

## Data Availability

The data presented in this study are available on request from the corresponding author.

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
