# Peer review of "Solid Oxide Fuel Cells with Magnetron Sputtered Single-Layer SDC and Multilayer SDC/YSZ/SDC Electrolytes"

_membranes, 2023, doi:10.3390/membranes13060585_

Round 1

Reviewer 1 Report

In this work entitled “Solid oxide fuel cells with magnetron sputtered single-layer SDC and multilayer SDC/YSZ/SDC electrolytes”, the electrochemical properties of anode-supported SOFCs with magnetron sputtered single-layer SDC and multilayer SDC/YSZ/SDC thin-film electrolyte with the YSZ blocking layer 0.5, 1 and 1.5 μm thick. The fuel cell with the multilayer SDC/YSZ/SDC electrolyte with the layer thicknesses of 3/1/1 µm has the maximum power density of 2263 and 1132 mW/cm2 at 800 and 650°C, respectively. In general, the work is of interest for the readers of the journal of Membranes. However, the manuscript should be carefully and thoroughly checked by the authors, especially concerning the cell power density and polarization data, and the following suggestions could be considered for the authors.

1.       In Figure 6. Temperature dependences of OCV of SOFCs with single-layer SDC electrolyte (5.5 μm

thick) and multilayer SDC/YSZ/SDC electrolyte with different thickness of YSZ layer, the unit of the voltage should be mV instead of V.

2.       In the section of “3.2. Fuel cell with multilayer SDC/YSZ/SDC electrolyte”, the cross-sectional SEM image of multilayer SDC/YSZ/SDC electrolyte after the SOFC tests could be included for a good comparison, which is of great interest for the readers.

3.       According to Table 3 and Table 4, the Polarization resistance for SDC/YSZ(1μm)/SDC SOFC is 7.2 Ω·cm2 at 500°C and the Ohmic resistance is 0.29 Ω·cm2, while the cell Power density of SDC/YSZ(1μm)/SDC cell is reported to be 201 mW at 500°C in Table 2. With such a large total resistance it is not possible to obtain such a power output of 201 mW at 500°C. Similarly for the values recorded at 600°C (for 1.08 Ω·cm2 total resistance it is not possible to achieve 794 mW power density at around 0.5 V in Figure 7b). Please carefully recheck all the data in the work and correct it. Otherwise the presented data are not reliable.

4.       Please recheck all figures caption, such as for Figure 2.

Moderate editing of English language required

Author Response

Responses to Reviewer #1’s comments:

  1. COMMENT: In Figure 6. Temperature dependences of OCV of SOFCs with single-layer SDC electrolyte (5.5 μm thick) and multilayer SDC/YSZ/SDC electrolyte with different thickness of YSZ layer, the unit of the voltage should be mV instead of V.

RESPONSE: The error has been corrected, the figure has been changed.

  1. COMMENT: In the section of “3.2. Fuel cell with multilayer SDC/YSZ/SDC electrolyte”, the cross-sectional SEM image of multilayer SDC/YSZ/SDC electrolyte after the SOFC tests could be included for a good comparison, which is of great interest for the readers.

RESPONSE: We did not indicate in the text that Fig. 5,a was obtained after testing the fuel cell. An appropriate mention is added to the text.

  1. COMMENT: According to Table 3 and Table 4, the Polarization resistance for SDC/YSZ(1μm)/SDC SOFC is 7.2 Ω·cm2 at 500°C and the Ohmic resistance is 0.29 Ω·cm2, while the cell Power density of SDC/YSZ(1μm)/SDC cell is reported to be 201 mW at 500°C in Table 2. With such a large total resistance it is not possible to obtain such a power output of 201 mW at 500°C. Similarly for the values recorded at 600°C (for 1.08 Ω·cm2 total resistance it is not possible to achieve 794 mW power density at around 0.5 V in Figure 7b). Please carefully recheck all the data in the work and correct it. Otherwise the presented data are not reliable.

RESPONSE: High values of polarization resistance are related to the measurement of impedance at open circuit voltage. This measurement does not accurately reflect the operation of the fuel cell. Therefore, as a rule, the impedance must be taken under load conditions. In this case, the polarization resistance decreases dramatically. Unfortunately, our impedance meter does not allow measuring impedance spectra under load conditions.

As a proof I present the results of measuring the I-V characteristics and impedance spectra in [Cho et al. High-performance thin film solid oxide fuel cells with scandia-stabilized zirconia (ScSZ) thin film electrolyte, International Journal of Hydrogen Energy,  40 (2015) 15704-15708]. In this paper, the maximum power density of the SOFC was measured to be 227 mW/cm2 (500°C) and 334 mW/cm2 (550°C). EIS measurements were also conducted under different applied cell voltages (OCV, 0.7 and 0.5 V) to clarify the cell's electrode polarization resistance and ohmic resistance characteristics (Fig. 4b). The ohmic resistance (the X-axis intercept point at high frequency) was somewhat similar for the two applied cell voltage conditions (0.340 Ohm*cm2 at 0.7 V, 0.256 Ohm*cm2 at 0.5 V at 500°C; Fig. 4b inset). The polarization resistance, was affected by the applied cell voltage (more than 10 Ohm*cm2 at OCV, 1.513 Ohm*cm2 at 0.7 V and 1.018 Ohm*cm2 at 0.5 V at 500°C). Thus, the values of power, ohmic and polarization resistance (measured at OCV) in our paper and Cho's paper have comparable values. When measured under load, the polarization resistance in Cho et al. paper decreases by an order of magnitude.

Fig. 4. (a) Polarization curve and power density of the TF-SOFC with the ScSZ thin film electrolyte at 500–550°C. (b) EIS results of the TF-SOFC [G.Y. Cho et al., High-performance thin film solid oxide fuel cells with scandia-stabilized zirconia (ScSZ) thin film electrolyte, International Journal of Hydrogen Energy,  40 (2015) 15704-15708. http://dx.doi.org/10.1016/j.ijhydene.2015.09.124].

  1. COMMENT: Please recheck all figures caption, such as for Figure 2.

RESPONSE: We checked the captions under the figures, but found no errors.

Reviewer 2 Report

I find that the Review entitled: "Solid oxide fuel cells with magnetron sputtered single-layer SDC and multilayer SDC/YSZ/SDC electrolytes" is very an interesting.

In this paper was presented the possibility of using magnetron sputtering was demonstrated for a successful fabrication of the thin film SDC electrolyte for IT-SOFCs. The paper compares the properties of anode-supported SOFCs with magnetron sputtered single-layer SDC and multilayer SDC/YSZ/SDC thin-film electrolyte with the YSZ blocking layer 0.5, 1 and 1.5 μm thick. Magnetron sputtering provided the formation of the defect-free electrolyte representing the cubic phase and having good adhesion to the substrate. The authors explained that the OCV of the fuel cell with the SDC electrolyte 5.5 μm thick, was about 0.8 V in the temperature range of 500 to 650°C, and which was then decreased to 0.77 V at 700°C. The highest SOFC performance was observed at 650°C, when its OCV and power were 0.805 V and 651 mW/cm2, respectively. In additionally, the formed was of the SDC electrolyte with the YSZ blocking layer improves the open circuit voltage up to 1.1 V and increases the maximum power density at the temperatures over 600°C. The authors showed that the optimal thickness of the YSZ blocking layer is 1 µm. At 650°C, the SOFC with the SDC/YSZ(1 μm)/SDC electrolyte demonstrated 1.1 V and 1132 mW/cm2, i.e., 1.5 times higher than manifested by the SOFC with the single-layer electrolyte. The highest performance was observed for the fuel cell with the multilayer electrolyte at 800°C, its OCV and power density were respectively 1.06 V and 2263 mW/cm2.

Although the authors stated that further research will be needed to test the long-term stability of multilayer electrolytes and their resistance to redox tests, I think that the idea is innovative and promising. I suggest the authors to continue their research in this direction.

Accordingly, I recommend accept in present form.

Author Response

Since Reviewer 2 made no comments, we do not respond. 

Round 2

Reviewer 1 Report

Accept in present form

Minor editing of English language required